Clocks at a snail pace: biological rhythms in terrestrial gastropods

Salvador Rodrigo Brincalepe 1 2
Tomotani Barbara Mizumo barbara.m.tomotani@uit.no 3
1 The Arctic University Museum of Norway, UiT - The Arctic University of Norway , Tromsø , Norway
2 Finnish Museum of Natural History, University of Helsinki, University of Helsinki , Helsinki , Finland
3 Department of Arctic and Marine Biology, UiT - The Arctic University of Norway , Tromsø , Norway
Garant Dany
Electronic publication date: 2024 Oct 29
Publication date: 2024
Volume: 12
Electronic Location ID: e18318
Received 2024 Mar 20; Accepted 2024 Sep 24
Copyright: ©2024 Salvador and Tomotani
Copyright year: 2024
Copyright holder: Salvador and Tomotani
License: This is an open access article distributed under the terms of the Creative Commons Attribution License, which permits unrestricted use, distribution, reproduction and adaptation in any medium and for any purpose provided that it is properly attributed. For attribution, the original author(s), title, publication source (PeerJ) and either DOI or URL of the article must be cited.
License URL: https://creativecommons.org/licenses/by/4.0/

Keywords: Biological clocks, Daily rhythms, Land snails, Systellommatophora, Stylommatophora, Aestivation, Hibernation, Annual rhythms

Funding: Velux Stiftung, Switzerland Proj. 1974 This project was funded by the Velux Stiftung, Switzerland (Proj. 1974). The funders had no role in study design, data collection and analysis, decision to publish, or preparation of the manuscript.

==============================
Biological rhythms are ubiquitous across the tree of life. Organisms must allocate their activities into moments of the day and of the season that will increase their probability of surviving and reproducing, which is done in the form of daily and annual rhythms. So far, the vast majority of studies on biological rhythms have focused on classical laboratory model species. Still, the use of non-model species is gaining traction, as part of an effort to achieve a more holistic understanding of clock/calendar mechanisms in the “real world” but this requires species that can be studied in both the lab and in nature. Terrestrial gastropods, i.e., land snails and slugs, have the potential to be exciting models for the study of biological rhythms in nature. Therefore, we provide a review of the research on biological rhythms in terrestrial gastropods, with a focus on ecology and evolution. We present the state of the art in the field while giving a historical perspective of the studies, exploring each of the main lineages of terrestrial gastropods. We also point out some interesting directions that future studies could take to fill some of the more urgent gaps in current knowledge. We hope that our contribution will renew interest in this area and spark novel projects.

Introduction

The Earth’s rotation around its axis and the sun creates challenges that recur periodically across the day and the year for living organisms. For example, daily cycles of light and darkness and yearly changes in daylength that are more or less pronounced depending on the latitude. Organisms must thus allocate their activities into moments of the day and of the season that will increase their probability of surviving and reproducing, which is done in the form of daily and annual rhythms (Enright, 1970; Helm et al., 2017).

Biological rhythms, in particular daily rhythms, are ubiquitous across the tree of life, from bacteria to multicellular eukaryotes. For example, organisms possess daily rhythms in behavioural (activity and rest), physiological (metabolic, hormonal, photosynthesis), and even cellular functions (Liao & Rust, 2021; Panda, Hogenesch & Kay, 2002). An important feature of such biological rhythms is the fact that they are not simply responses to the changing environmental conditions but are endogenously generated by a circadian clock. Thus, even in the absence of any environmental information, daily rhythms persist with their own properties (Pittendrigh & Daan, 1976; Aschoff, 1981). Once in natural conditions, circadian clocks are synchronized to the environment, and thus the rhythms acquire a 24-hour period and establish a stable phase relationship to the environmental cycle, with the light/dark cycle being the most important environmental cue for the synchronization of daily clocks. Similarly, annual rhythms are synchronized by the variation in photoperiod (i.e., day length, or the interval of time each day an organism receives light) but different species also have specific supplementary cues for the optimal timing of seasonal events (Visser et al., 2010).

The majority of studies on biological rhythms have focused on classical model species, in particular those that are already established as laboratory models (Kronfeld-Schor, Bloch & Schwartz, 2013). Still, the use of non-model species has increasingly gained attention in Chronobiology, allowing a more holistic understanding of the clock/calendar mechanisms and their significance in the “real world” (Kronfeld-Schor, Bloch & Schwartz, 2013; Helm et al., 2017). Ideally, a species should be suitable for both field and laboratory investigations, although this is not always easy or straightforward (Schwartz, Helm & Gerkema, 2017). This approach allows rhythms to be studied (1) in a natural context, where their ecological and evolutionary significance can be assessed, and (2) in the lab, where anatomical, physiological, and molecular aspects can be investigated. In this review we propose that terrestrial gastropods can be excellent study systems, albeit still understudied.

Mollusca is the second most diverse animal phylum, with over 80,000 valid species and more than 100,000 estimated to be still unknown (Rosenberg, 2014; Bouchet et al., 2016). Like other organisms, molluscs also present daily and annual rhythms but particularities of their anatomy and ecology create specific challenges for the group. Terrestrial gastropods (land snails and slugs), for example, are constantly challenged by the availability of moisture in their environment and thus need to allocate their activity to moments of the day when desiccation risk is lowest (i.e., the night; Cook, 2001). They also possess abilities to deal with extended and predictable periods of cold temperatures in winter or with dry seasons, being able to become dormant for very long periods (Cook, 2001). They must not only predict such changes in the environment but also synchronize their activities with conspecifics during the breeding season, thus breaking dormancy in a synchronous manner, which suggests the involvement of endogenous biological clocks in the regulation of those activities (Cook, 2001).

Terrestrial gastropods thus possess qualities that make them exciting models for the study of biological rhythms in nature, as we will show in this review. They are a rich biodiverse group including lineages with different evolutionary histories (Vermeij & Watson-Zink, 2022); they are widespread, abundant, and relatively easy to track and to keep in captivity for experimental protocols; they have a centralized and peculiarly arranged nervous system; and they have pronounced annual rhythms, including hibernation and aestivation (and sometimes both), with a large variation in strategies due to the range of habitats they live in. Not only species comparisons are possible, but some species such as the garden snail (Cornu aspersum) have also colonized several different habitats allowing for studies of local adaptation to a range of environments. Finally, land snails also have economic and public health relevance due to their use as food items (and, more recently, as pets), their impact on crops, and because some act as intermediate hosts of parasites (Machado et al., 2023; Salvador et al., 2024).

But despite the early historical importance of terrestrial gastropods in the field of Chronobiology (Newell, 1968; Lewis, 1969a), they have not received much attention since. Molluscs (like many other invertebrates) suffer from a lack of studies on par with their diversity, ecological importance, and prominence across most types of environments and habitats. That goes from general biology (Davison & Neiman, 2021) to Chronobiology.

Here we provide a review of the research on biological rhythms in terrestrial gastropods, focusing largely on the Stylommatophora, i.e., the group for which daily and annual rhythms are better described. We present the state of the art in the field while giving a historical perspective of the studies and also point out some directions that future studies could take to fill some of the more urgent gaps in current knowledge.

Survey Methodology

We conducted an extensive literature review of studies on biological rhythms, both daily and annual, in land snails and slugs. We did not restrict ourselves to chronobiological or experimental studies but also included ecological, natural history, and other types of studies, particularly if an experimental component was present. However, we had to draw the line somewhere, so studies focusing only on the description of life histories or phenology (including seasonal variation in metabolism, hormones, etc.) were not included here.

As expected, most studies focus on European species, so we strived to bring in as much information as we possibly could from studies and species in the Global South. We are aware that we cannot possibly cover all regions and languages and that our efforts might fall short in some cases. So, if the reader finds out that we missed some studies, we urge them to contact us.

The literature in this area of study is not vast, which allowed us to be very artisanal and go deep in our searches. We conducted our searches through Google Scholar using the following terms: Gastropod(a), Snail or Slug, alongside Rhythm, Daily, Seasonal(ity), Clock, Circadian, Circannual, Activity, Aestivation, Hibernation or Dormancy. Searches for these terms were conducted in English, Spanish, Portuguese, French, and German. A VPN service was used to switch the location of the connection when conducting the searches, using, respectively, the following countries: UK, Mexico, Brazil, France, Germany. Furthermore, every time an article was read, all its references were checked in search of entries not still found during our previous searches.

A list of the studies reviewed in the present work can be found in Supplementary Material 1, which contains three spreadsheets: one with daily rhythms, one with annual rhythms, and one with the list of references. The spreadsheets in Supplementary Material 1 also have information on the model species of each study and a summary of each study’s main findings.

In total, we have reviewed 111 studies dealing with 46 different species of terrestrial gastropods (43 of which are Stylommatophora, while the remaining three are Veronicellidae). As we had to maintain our review focused, not all those studies are reported in the text that follows. Of the 111 studies, 66 are referred to in this text (some in greater detail, others only to exemplify specific points); the remaining 45 studies are present only in Supplementary Material 1. Therefore, we urge the reader to refer to Supplementary Material 1 for a more complete overview of the studies on a given species or topic of interest.

Note on the usage of terms

Irregular use of some terminology is common in the literature, particularly across disciplines. For instance, the term ‘period’ is often used as a synonym of ‘phase’ in non-chronobiological studies. Herein, when discussing the findings of such papers, we have “updated” those terms to their usage in the discipline of Chronobiology. A glossary of terms in Chronobiology (jargons) is presented further below.

Similarly, molluscan classification has changed quite significantly in the past decades. Species have sometimes been moved to other genera or synonymized with other species. Thus, the scientific names of species used herein have also been updated according to current classification and taxonomic usage (MolluscaBase, 2024). Consequently, the names used in the present paper (including in the Supplementary Material) in several cases will differ from those in the original studies referenced here.

Finally, there is one particular group of problematic terms that deserve further attention. In the literature, it is not uncommon for terms such as ‘hibernation’, ‘aestivation’, ‘dormancy’, and even ‘diapause’, to be used interchangeably. There are also concerns regarding the stricter definitions of these terms. Thus, to avoid confusion, herein we define our terminology as follows.

The term ‘hibernation’ in a strict sense cannot be applied to ectothermic animals because it involves a programmed decrease in body temperature. Thus, the correct term for terrestrial gastropods would be ‘dormancy’ or ‘winter dormancy’. Nevertheless, the usage of ‘hibernation’ has become solidified in molluscan research and literature as an analogue of the “true hibernation” observed in warm-blooded animals (Barker, 2001; Cook, 2001). Thus, herein we use ‘hibernation’ to refer to a physiologically programmed seasonal process that takes place during the colder months, in which metabolism is greatly reduced.

Aestivation is the dormancy during unfavourable conditions (e.g., dryness, heat) and the term is applied in the literature for both short-term (immediate) processes and long-term (with a seasonal component) processes (see ‘Annual and circannual rhythms’ below for a full explanation). We maintain here the use of the same term, ‘aestivation’, for both cases such as it is applied in the literature; however, we always make clear to which of those two processes we are referring to.

Finally, in the literature sometimes the term ‘diapause’ is used for snails, and it can refer to either hibernation or aestivation. Nevertheless, this term is historically connected to arthropods (Chapman, Simpson & Douglas, 2013), and thus, it is not applicable to terrestrial gastropods.

Glossary

When reaching across disciplines, we can reduce the risk of misinterpretation by defining the key terms used (Stankowski et al., 2024). Throughout this article, we use terms that are either Chronobiology jargons or that are more general terms that have narrower field-specific definitions within Chronobiology. Thus, we provide here a glossary with a brief definition of each term that the reader can refer to. More detailed explanations of these terms can be found in Dunlap, Loros & De Coursey (2004), Schwartz, Helm & Gerkema (2017), and the Dictionary of Circadian Physiology (http://www.circadian.org/dictionary.html).

• Biological clock: Endogenous system that generates biological rhythms without an external periodic input.

• Biological rhythm: A biological function that repeats itself with a defined periodicity (e.g., locomotor activity and rest).

• Circadian: Scientific term formed from Latin, meaning “circa one day” (i.e., ∼24 h). Rhythms with a period of approximately 24 h. For a rhythm to be considered circadian it must be endogenously generated, entrainable by a Zeitgeber and temperature compensated.

• Circannual: Scientific term formed from Latin, meaning “circa one year” (i.e., ∼365 days). Rhythms with a period of approximately 365 days. For a rhythm to be considered circannual it must be endogenously generated and entrainable by a Zeitgeber (see Hazlerigg et al., 2023 for various examples).

• Diurnal: An organism that has its activity during the day, i.e., in light conditions (or during subjective day when in constant conditions).

• Endogenous: That originates within the organism.

• Entrainment: The synchronization of a self-sustained rhythm (e.g., the endogenous circadian rhythm) by a forcing cycle (the Zeitgeber, e.g., the daily cycle of light and darkness), resulting in both to run with the same period and a stable phase relationship.

• Free-running: State of a self-sustaining rhythm in constant conditions (LL or DD) and not exposed to any external time cues that would affect their period. In such conditions, rhythms exhibit their endogenous (not entrained) period. The endogenous period length of a rhythm is also known as ‘tau’ (τ).

• Masking: A change in the expression of an organism’s rhythm, which, in contrast to entrainment, does not involve a change in the period or phase of the endogenous rhythm.

• Nocturnal: An organism that has its activity during the night, i.e., in dark conditions (or during subjective night when in constant conditions).

• Period: The amount of time it takes a cyclical process to complete. For instance, from the start of the activity of a given day to the start of the activity the following day.

• Phase: Each moment of a cycle, an instantaneous state of an oscillation, a reference point (e.g., start of the activity).

• Photoperiod: Also known as ‘day length’. The duration of the light phase in a light-dark cycle. For instance, a condition of LD14:10 has a longer photoperiod than a condition of LD10:14 (though both cycles are 24 h long, i.e., both have a 24 h period).

• Skeleton photoperiod: A light-dark cycle lacking a long period of light (i.e., the “day phase”); instead, short bouts of light mark the beginning and the end of the day phase.

• Tau (τ): The endogenous period length of a rhythm.

• Temperature compensation: A property of a biological process where the rate of the process is preserved despite changes in the surrounding temperature.

• Zeitgeber: From the German, “time-giver”. A cue, external to the organism, that can synchronize a rhythm. One important aspect is that Zeitgebers do not induce a rhythm, they determine their period length and phase.

Who Are the Terrestrial Gastropods?

Even though terrestrial gastropods are usually considered a single group by most, there are several different and phylogenetically unrelated lineages of gastropods that independently colonised the land (Barker, 2001; Vermeij & Watson-Zink, 2022). That is, the group is not monophyletic. There are about 25,000 living species of terrestrial gastropods (Rosenberg, 2014), belonging to the three largest subclasses of Gastropoda (Neritimorpha, Caenogastropoda, and Heterobranchia); thus, members of one lineage are only remotely related to those of another lineage (Fig. 1). The terrestrial forms in the first two subclasses are usually grouped as “operculate snails”, as they have a calcareous operculum that closes the aperture of the shell when the animal retracts its soft tissues inside.

The Heterobranchia contains marine and freshwater animals, as well as some distinct branches that colonised land independently (Fig. 1): members of the family Amphibolidae (which are also operculate) and members of the superorder Eupulmonata, whose lineage lost the operculum and developed a lung to breathe air. Members of Eupulmonata are informally known as ‘pulmonates’.

Eupulmonata contains three orders (Fig. 1): the marine Ellobiida, in which family Ellobiidae contains some terrestrial forms; the Systellommatophora, which contains the Onchidiidae slugs with both marine and terrestrial forms, as well as the terrestrial Veronicelloidea slugs (leatherleaf slugs); and the Stylommatophora, which are entirely terrestrial and contain the vast majority (ca. 80%) of land snail and slug biodiversity (Rosenberg, 2014).

Figure 1 Schematic phylogenetic tree of Gastropoda (after Ponder, Lindberg & Ponder, 2020) showing the main branches with terrestrial representatives.

Photos of terrestrial gastropods are by Olivier Gargominy (Ellobiida; source: MNHN & OFB, 2003-2024, CC BY SA 4.0) or from our own works (other taxa; source: Salvador et al., 2018, CC BY SA 4.0).

Therefore, the Stylommatophora are typically what both academic and popular articles mean when they use generic terms such as “terrestrial gastropods” or “land snails/slugs”. Considering their numbers and prevalence in the environment (particularly in the temperate Global North where most research has historically taken place), the vast majority of studies pertain to stylommatophoran gastropods. Thus, they will be the main taxon discussed herein, though we present data on other groups when available.

Considering the paraphyly of terrestrial gastropods and their multifaceted evolution, we organized our review discussing each of the main taxonomic groups in turn. We start with Stylommatophora, followed by Ellobiida, Systellommatophora, and the operculate land snails. We believe that evolutionary history and phylogenetic relationships are the most important aspects to be considered when contrasting taxa in comparative studies; after all, and to paraphrase a famous quote, meaningful conclusions can only be drawn under the light of evolution. Therefore, each lineage must be understood within its own history, paying attention to its phylogenetic relationships with other taxa and, importantly, considering whether observed traits of the organism are plesiomorphic (“ancestral”) or apomorphic (“derived”). It is also worthwhile to remember that the Eupulmonata diverged from other Heterobranchia (including the freshwater Hygrophila; Fig. 1) during the Mesozoic, while other lineages of gastropods (including the operculate terrestrial snails) mostly diverged from each other even earlier. Thus, it is not useful or correct to lump an entire category of organisms (e.g., terrestrial gastropods, all gastropods, or even molluscs in general) for comparisons with other taxa as if they were a single entity. That practice can lead to comparisons largely devoid of meaning and, more concerningly, to mistaken conclusions. On the other hand, if taken into consideration, the paraphyly of terrestrial gastropods makes these animals very interesting models for studying parallel evolution of traits (Vermeij & Watson-Zink, 2022), including daily and annual rhythms.

Finally, information on non-terrestrial Heterobranchia is sometimes needed for a full understanding, because terrestrial stylommatophorans share several traits with them. Marine Heterobranchia, in particular, have been historically important in Chronobiology and those studies have to a great extent guided the research later done on their terrestrial relatives (Attia, 2004). Notably, the first retinal clocks to ever be identified were studied in these animals using the California sea hare (Aplysia californica) and the California bubble snail (Bulla gouldiana) as model systems (Herzog & Block, 1999). Such background information will be provided when needed in the text that follows, but a more detailed overview of marine and freshwater heterobranchs can be found in Supplementary Material 2.

Biological Rhythms in the Stylommatophora

The Stylommatophora are the main source of biodiversity within Eupulmonata and land snails in general. Stylommatophorans are distributed worldwide and contain snails, slugs, and semi-slugs, including “well-known” species such as the Garden Snail, the Roman Snail (or escargot), the Leopard Slug, and the Giant African Snail.

Daily and circadian rhythms

Circadian rhythms are generated by an endogenous circadian clock and have an intrinsic period of approximately 24 h, named ‘tau’ (τ). External environmental cues (Zeitgebers) synchronize the circadian clock, resulting in a daily rhythm that matches the 24 h period of the environmental cycle. There are several conditions for a rhythm to be called “circadian” (Johnson et al., 2004), namely: endogenous generation, entrainability by a Zeitgeber and temperature compensation (see Glossary above). Thus, behaviours and physiological traits periodically recurring every 24 h in natural settings are better called daily rhythms or day/night rhythms. We use “circadian rhythms” and “circadian clocks” in this review with the caveat that only the first two criteria have been demonstrated in land snails and slugs.

However, organisms are also able to respond behaviourally to cues, without the involvement of the clock (masking). For instance, high temperatures may inhibit the activity, without a change in the synchronized clock.

Studying rhythms requires monitoring the animals throughout the day and night, which, despite challenges, can be carried out in field conditions. By contrast, when testing which cues (light/temperature) are important to synchronize rhythms, laboratory manipulations exposing the animals to one cue at a time are needed. Finally, when measuring the animals’ intrinsic rhythm generated by their internal clock, it is necessary to remove the cues that they use to synchronize their rhythms, so they only express their endogenous rhythms. Thus, this requires keeping animals in constant environmental conditions (e.g., constant darkness, constant temperature) in specialized laboratory settings.

Behavioural activity rhythms

Activity rhythms of feeding and locomotion (referred to just as “activity” in this review, as they are usually studied together) are a commonly studied clock output that at the same time gives an indication of the ecological relevance of the clock and can serve as a proxy for studying its endogenous properties. While studies of activity patterns in stylommatophoran gastropods can be traced back to the 19th century, it was only in the second half of the 20th century that researchers started to experimentally manipulate light, humidity, and temperature to assess their effect on activity rhythms. This trend, of course, ran together with the beginnings of the field of Chronobiology in the 1950s and 1960s. Initially, it was difficult for researchers to disentangle the effects of light, temperature, and humidity on the activities of snails and slugs (e.g., Dainton, 1954a; Dainton, 1954b; Daxl, 1969). That changed when new experimental methodologies and protocols started to emerge in chronobiological research and to be applied to the study of molluscs (e.g., Henne, 1963).

There are numerous studies describing the activity patterns of stylommatophoran snails and slugs. It is well-known that nearly all studied species are nocturnal and often also active during dusk and/or dawn (e.g., Cook, 2001; Speiser, 2001) (Supplementary Material 1). This is understood as a strategy for saving water and avoiding desiccation, but few studies looked into activity as a daily rhythm, that is, they were not conducted within a rhythms-based framework considering the mechanisms responsible for the observed patterns (Cook, 2001).

The duration of daily activity of snails and slugs accompanies natural changes in photoperiod. For instance, nocturnal species will stay active for longer during months when nights are longer (e.g., Bailey, 1975; Rollo, 1982; Ford & Cook, 1994) and might expand their activity to daytime when nights are shorter (e.g., Staikou, Lazaridou-Dimitriadou & Kattoulas, 1989). Environmental variables such as relative humidity (particularly of the soil), rain, temperature, and wind speed can all have a direct effect on the onset and duration of activity, even during times of the day outside “regular” active hours (Cook, 2001; Attia, 2004). Some species that are commonly active during twilight hours typically extend their activity into the early hours of the morning, when temperatures are still low and relative humidity is higher. Furthermore, under the same conditions of light and darkness, conspecific animals vary their daily activity depending on the temperature (e.g., Henne, 1963; Cameron, 1970). Historically, that has led to confusion over whether these animals had endogenously generated rhythms or not (e.g., Tischler, 1974). However, it has been later demonstrated that snails and slugs, like most animals, have self-sustained endogenous rhythms synchronized to light/dark cycles (see ‘Circadian rhythms and circadian clocks’).

Not all terrestrial gastropods are nocturnal; there are a few species known to deviate from the norm. The dot snail Punctum pygmaeum was shown to be active in the lab during intervals of a few hours scattered throughout the whole day (Baur & Baur, 1988). The dot snail is a microgastropod that lives in the leaf litter, leading Baur & Baur (1988) to hypothesize that activity scattered throughout the whole day would be a trait of snails inhabiting such microhabitats. That seems to be the case for other microsnails, belonging to families not immediately related to Punctum, observed in the field (Boag, 1985) and in the lab (Pilate et al., 2014b). However, whether they present a circadian rhythm under constant conditions has not been tested. More unusually, the Pacific banana slug Ariolimax columbianus is active during the day during spring and fall and more crepuscular during summer (Richter, 1976), but the reasons for this remain uncertain. One interesting and still unknown question is what the daily activity rhythms of snails and slugs look like when living under extreme conditions such as in aphotic areas in caves or in the arctic summer when there are 24 h of daylight.

Physiological rhythms

Although the majority of studies have focused on activity (locomotor or feeding) rhythms, physiological processes also occur rhythmically (Supplementary Material 1). For instance, daily rhythms of heart rate and chemosensitivity were reported for the Roman snail Helix pomatia (respectively, Wünnenberg, 1991; Voss, Kottowski & Wünnenberg, 1997), while an central nervous system activity rhythm was reported for Cryptozona ligulata from India (Reddy et al., 1978). As expected for nocturnal animals, the snails had a higher heart rate (Wünnenberg, 1991), maximum neural responses (Voss, Kottowski & Wünnenberg, 1997), and higher electrical activity in the central nervous system (Reddy et al., 1978) during the dark phase. Wünnenberg (1991) also showed that measurement of heart rate is a suitable methodology for studying rhythms in dormant animals, i.e., during hibernation or aestivation (see ‘Annual and circannual rhythms’ below).

Similarly, the daily rhythms of some enzymes and other molecules have also been studied to some extent. For instance, in the megasnail Megalobulimus abbreviatus from Brazil, the antioxidant enzymes glutathione peroxidase (GPx) and superoxide dismutase (SOD) have their peak activity in the CNS and mantle in the middle of the light phase and thus, do not match the snail’s daily rhythm of oxygen consumption (Babini Junior et al., 2007). There are further examples of metabolic studies such as the one above (see Supplementary Material 1), but this is still a nascent line of research and interpretations of the findings in relation to the organisms’ biology are not always straightforward.

Furthermore, given its involvement in vertebrate circadian rhythms, the hormone melatonin has also been studied in snails. Blanc et al. (2003) showed that melatonin is present in the nervous system of the garden snail Cornu aspersum, and they detected a small peak in the cerebroid ganglions at the end of the night. Those authors also described a daily rhythm of 5-methoxytryptophol (5-ML) in the ocular tentacles, with low concentration in the middle of the light phase and high concentration during the dark phase. 5-ML has been described to be synthetized in the pineal gland during the day in several vertebrates and in the retina of some non-mammalian vertebrates. It is described to be regulated by photic stimuli and cycle out of phase with melatonin rhythm (Zawilska et al., 2003). Thus, the results obtained by Blanc et al. (2003) contrast with those for vertebrates, with snails having a peak during the night. The function of 5-MT may be similar to that of melatonin but further investigations in invertebrates are lacking (Ouzir et al., 2013); thus, the results of Blanc et al. (2003) remain unexplained.

Circadian rhythms and circadian clocks

As discussed above, in comparison to the studies of daily activity/physiological rhythms, studies on circadian rhythms, clocks and the response of clocks to light are much scarcer. In pioneering studies, Newell (1968) and Lewis (1969a) demonstrated that slugs (respectively, the grey field slug Deroceras reticulatum and the black slug Arion ater) synchronized their activity to light cycles and not temperature. Lewis (1969b) demonstrated that there was an endogenous circadian activity rhythm controlled by light/dark cycles by keeping the animals in constant darkness for at least 2 weeks. Subsequently, the same was demonstrated for snails and other species of slugs (Supplementary Material 1). The clock of snails and slugs began to be better understood after (1) their clocks’ intrinsic period (τ) started to be investigated, (2) experimental manipulations of light/dark cycles and light pulses were done, (3) it became clear that temperature and humidity were masking the rhythms in nature, and (4) neurobiological studies of the clock began (though mostly in other Heterobranchia; see also Supplementary Material 2).

Endogenous rhythms.

As observed in other species, when in constant conditions, snails show a species-specific endogenous period length (τ) measured via their locomotor and/or feeding activity rhythm; for example, 23.6<τ<24.6 h in Limax maximus (Sokolove et al., 1977), and τ>24 h in both Limax ecarinatus (Ford & Cook, 1987; Cook & Ford, 1989) and Cornu aspersum (Lorvelec et al., 1991). However as observed in other groups, τ is a labile trait and varies depending on individual, environmental and/or experimental conditions (Aschoff, 1979; Pittendrigh & Daan, 1976). For example, it is known that the exact length of τ varies from one individual to the next within the same population of both snails and slugs (Sokolove et al., 1977; Lorvelec et al., 1991). Moreover, τ responds to changes in lab conditions, being shorter in LL than DD in Limax maximus (Sokolove et al., 1977). The latter is an interesting result because it contradicts one of the so-called “Aschoff’s rules”, which states that in nocturnal animals τ becomes longer in LL relative to DD (Aschoff, 1960). This could indicate that terrestrial gastropods may be exceptions to the rule, but we are still lacking similar experiments in other snails or slug species for testing this.

The strength of the rhythmicity is also variable, for example, the endogenous locomotor activity rhythm of young snails (Cornu aspersum) is weaker in comparison to adults in constant dim light conditions, but it is still entrained by LD, as in the adults (Blanc, 1993). It has been reported that Helix pomatia becomes behaviourally arrhythmic when exposed to constant LL or DD conditions, but that was later raised as potentially a methodological issue (e.g., Gelderloos, 1979) considering that other stylommatophorans, including closely-related species, are known to free-run in both LL and DD (e.g., Sokolove et al., 1977; Bailey & Lazaridou-Dimitriadou, 1986; Flari & Lazaridou-Dimitriadou, 1995a). However, the possibility of arrhythmicity in some individuals or populations raises an interesting point, as it is not known whether snails/slugs exposed to natural constant darkness in caves or constant light above the Arctic Circle are naturally rhythmic or arrhythmic, like in some vertebrates (Beale et al., 2013; Hazlerigg et al., 2023).

Light treatment.

Many studies tested the effect of light as a clock synchronizer (Zeitgeber) in snails and slugs. Besides the standard studies on activity patterns using light and dark cycles, some researchers have used more specific protocols such as light pulses, use of light colours, among others.

Flari & Lazaridou-Dimitriadou (1995b) exposed specimens of Helix lucorum, a nocturnal European species, to varying skeleton photoperiods. Their protocol consisted of two 1 h long light pulses in each 24 h cycle; the first light pulse was “movable”, i.e., given at a variable time along the 24 h dark phase, and the second one was fixed, i.e., always given at the last hour of the dark phase. Thus, each of their 23 treatments had a longer and a shorter interval between the two pulses. The snails were able to synchronize to most treatments, and the response to light of their clocks followed the same properties as insects and needed a dark phase of at least ten hours to remain nocturnal (Flari & Lazaridou-Dimitriadou, 1995a). This species does not experience too short nights in nature (Staikou, Lazaridou-Dimitriadou & Kattoulas, 1989), but under laboratory conditions of photoperiod regimes of 9 or 7 h of dark, the animals extend up to 30% of their activity into the light phase (Lazaridou-Dimitriadou & Bailey, 1991).

Light pulses and skeleton photoperiods were also shown to entrain the activity rhythm of the slug Limacus ecarinatus (Ford & Cook, 1988). Snails and slugs were shown to synchronize to light/dark cycles shorter than 24 h periods, such as 20 (i.e., Lazaridou-Dimitriadou & Bailey, 1991) or even 19 h (i.e., Ford & Cook, 1987) and it would be interesting to follow-up such studies via tests of the limits of entrainment and creating phase response curves (Pittendrigh & Daan, 1976) for terrestrial gastropods.

Light has also been shown to have a direct non-clock effect on the activity rhythm (masking). For example, some nocturnal species show an additional locomotor activity peak in the morning that has been hypothesized to have an exogenous origin in Limax maximus and Cornu aspersum (Sokolove et al., 1977; Bailey, 1981). Sokolove et al. (1977) argued that this morning “lights-on” burst of locomotor activity was a laboratory artifact, a light-avoidance escape response. Other studies, however, do show that both the nocturnal locomotor activity and the morning peak are endogenous, e.g., Helix lucorum (Bailey & Lazaridou-Dimitriadou, 1986). Bailey & Lazaridou-Dimitriadou (1986) further hypothesized that the locomotor activity rhythm would be controlled by two oscillators as proposed for insects, because the phase relationship of the two activity components (morning and evening) would shift in constant conditions, drifting apart from each other. Similarly, in a study with the slug Limacus ecarinatus, Ford & Cook (1987) showed that locomotor and feeding daily rhythms have different periods (τ) in constant conditions, which also led to the suggestion of control via two oscillators.

Finally, when it comes to light spectra, tests with the slugs Deroceras reticulatum and Limax maximus suggested that these two species are synchronized equally by red and white light (Dainton, 1954b; Beiswanger, Sokolove & Prior, 1981). In addition, in studies with Limax maximus, eyes were required for synchronization to red light but not white, suggesting the involvement of extraocular photoreceptors (see below) that are relatively insensitive to longer wavelengths (Beiswanger, Sokolove & Prior, 1981; Sokolove, Kirgan & Tarr, 1981a).

Photoreceptors and the clock.

In classical studies with marine Heterobranchia, the eyes were identified as important for the synchronization of daily locomotor activity rhythms (Supplementary Material 2), as the retinal cells harbour a circadian clock (Block & Colwell, 2014). Because the traditional test for finding the location of the clock involved the surgical ablation of eyes, the same method was thus similarly used in terrestrial gastropods. In the slugs Limax maximus and Limacus flavus (Beiswanger, Sokolove & Prior, 1981), and the snails Cornu aspersum and Achatina fulica (Takeda & Ozaki, 1986; Attia, 1996), eye ablation did not affect the maintenance of rhythmicity or entrainment to LD. This implied the existence of extraocular clocks and photoreceptors in these animals, similar to what was later shown for marine Heterobranchia, marine Systellommatophora, and freshwater Hygrophila (see Supplementary Material 2).

The clock of terrestrial gastropods was proposed to be located entirely within the central nervous system, capable of being entrained by both ocular and extraocular photoreceptors (Beiswanger, Sokolove & Prior, 1981; Attia, 2004). The ablation of one cerebral ganglion in eyeless Cornu aspersum modified the free-running period (τ), suggesting the involvement of the ganglia in the control of the locomotor rhythms (Attia, 1996; Attia, 2004). Attia (1996), based on histological data, suggested that the mesocerebral cells could be involved in the process. These findings, however, remain largely unreproduced and untested so far.

Extraocular photoreceptors were demonstrated to be present in the cerebral and parietal ganglia of stylommatophoran snails and slugs (e.g., Pašić et al., 1977; Beiswanger, Sokolove & Prior, 1981; Pašić & Kartelija, 1990; Wünnenberg, 1994; Attia, 1996; Attia, 2004). Based on experiments in marine heterobranchs with ablated eyes, Newcomb et al. (2014) hypothesized that both ocular and extraocular photoreceptors would be involved in photoentrainment, but the former would have the most influence on rhythm expression. In stylommatophorans, however, the contributions of each type of photoreceptor to rhythm expression remains unknown, but a similar scenario to that proposed by Newcomb et al. (2014) is expected.

Finally, the extraocular receptors were shown to have a different sensitivity to light wavelengths in Limax maximus, as eyeless slugs did not entrain to LD cycles of wavelengths greater than 600 nm but did entrain to white (i.e., they are insensitive to red light; Beiswanger, Sokolove & Prior, 1981; Sokolove, Kirgan & Tarr, 1981a), while intact slugs entrain to both white and red. While this is within expectations and similar to what has been observed in Aplysia sea hares (Block, Hudson & Lickey, 1974; Supplementary Material 2), it is still an interesting find considering that red light seems to be important for the synchronization of terrestrial gastropods (Dainton, 1954b). This implies that the extraocular photoreceptors could be a plesiomorphy shared with marine Heterobranchia, and offers support to the claim that ocular photoreceptors (capable of being synchronized by red light, an important trait for a snail/slug) would have the largest influence on rhythmicity (Beiswanger, Sokolove & Prior, 1981). If confirmed, the ocular red-light sensitivity could therefore be an innovation (apomorphy) of either Eupulmonata or Stylommatophora.

Obvious venues for future studies in terrestrial gastropods would include detailed studies on the clock, both on the cellular (e.g., “clock neurons”) and molecular (e.g., clock genes, DNA-protein interactions) levels. This would not only provide a fuller picture for land snails/slugs but also allow comparisons to marine heterobranchs and freshwater Hygrophila, in which these aspects have been investigated in more detail (Block & Colwell, 2014; Hamanaka, Hasebe & Shiga, 2023; Supplementary Material 2).

Annual and circannual rhythms

Land snails have up to three major annual events depending on the species and the environment where they live: reproduction (and juvenile development), hibernation, and aestivation. Similarly to circadian rhythms of locomotor and feeding activity, endogenous circannual rhythms of reproduction with a period of approximately 365 days which persist in constant conditions have also been described for land snails. Also, similarly to daily rhythms, light play an important role in synchronizing annual rhythms, with the variation in daylength (photoperiod) being the major cue in most organisms (Numata & Udaka, 2010). Thus, experiments have been conducted with land snails testing the role of photoperiod of as a cue for their annual rhythms.

Most studies done in stylommatophorans so far, however, focused on life history traits (particularly reproduction), including descriptive and experimental studies, as well as seasonal changes in metabolism and in the reproductive system (Numata & Udaka, 2010; Supplementary Material 1). Experiments also focused on how environmental factors impacts (mask) behaviours (e.g., relative humidity, substrate type) without considering the annual rhythm per se. Notably, a great deal of attention has been (somewhat understandably) paid to anthropochorous, invasive or pest species (Supplementary Material 1). We did not include studies describing phenology in this review as they would go beyond the scope (and length). However, we would like to acknowledge the relevance of the studies characterizing timing of reproduction, hibernation, and aestivation in nature, which are valuable assets for any further ecological or evolutionary studies of annual rhythms in snails.

Circannual rhythms

To the best of our knowledge studies testing the endogenous nature of annual rhythms have only been done for reproductive traits. The reproductive cycle in species of snails and slugs was shown to persist under constant conditions (Bailey, 1981) and to be synchronized by photoperiodic cues (Gomot, 1990, see the following section). The free-running periods (τ) of the reproductive cycle of studied species so far (Limacus flavus and Cornu aspersum) are shorter than one year (ca. 15–30 days shorter for L. flavus, though the period could not be reliably established; Segal , 1960; Bailey, 1981). Notably, the report of Segal (1960) was the first demonstration of a circannual rhythm in molluscs, in which populations of L. flavus were kept under constant conditions of photoperiod (LD 11:13) and temperature for three years. Curiously, the work of Segal (1960) has often been forgotten in the literature, which is very likely due to its unique publication as a commentary on a study about periodicity in avian reproduction (Marshall, 1960). We do not discard that hibernation could also be a self-sustained circannual rhythm analogous to those described for mammals (Pengelley, Aloia & Barnes, 1978) but this remains to be tested.

Photoperiod and other seasonal cues

When studying photoperiodism or the response of organisms to changes in daylength, one interest in Chronobiology is to understand how animals can measure changes in photoperiod (photoperiodic time measurement). In land snails, studies (described below) have been limited to the response of seasonal events to natural and manipulated photoperiods, while the formal properties and molecular basis of photoperiodic measurement remain understudied. Similarly to the case of circadian rhythms, the few existing studies in gastropods looking at molecular and anatomical basis of photoperiodism have focused on the freshwater Lymnaea stagnalis (see Hamanaka, Hasebe & Shiga, 2023 for a review), which is a less seasonal and more opportunistic species (Fodor et al., 2020).

Reproduction and development.

As described above, photoperiodic cues play a role in synchronizing the annual rhythms of reproduction in land snails. In nature, the transition of short days to long days is thought to induce maturation of the male but not female reproductive system of slugs (Limax maximus); the female system matured irrespective of photoperiod (reminding that stylommatophorans are hermaphrodites) (Sokolove & McCrone, 1978). Once sexual maturity is fully initiated, though, subsequent steps following gonadal growth or shrinkage and development of other sexual organs are likely taking place irrespective of the photoperiod experienced by the animal (Sokolove & McCrone, 1978). Further experiments showed photoperiodic control of the secretion of maturation-inducing factors by the cerebral ganglia (McCrone, Van Minnen & Sokolove, 1981; McCrone & Sokolove, 1986). In Ambigolimax valentianus, in which male and female maturation occurs simultaneously (or nearly so), photoperiod affects both (Udaka & Numata, 2008).

It has been experimentally shown that exposure to longer photoperiods (i.e., with a longer light phase) can stimulate growth and sexual maturation of both snails (Cornu aspersum) and slugs (Ambigolimax valentianus, Limax maximus) (e.g., Sokolove & McCrone, 1978; Gomot, Enée & Laurent, 1982; Hommay et al., 2001; Wayne, 2001). Longer photoperiods can also stimulate spermatogenesis in snails and slugs (Henderson & Pelluet, 1960; Melrose, O’Neill & Sokolove, 1983; Sokolove et al., 1984; Gomot & Griffond, 1987; Wayne, 2001) and oviposition in Cornu aspersum (even in individuals with surgically removed eyes; Stephens & Stephens, 1966); some of these results, however, are ambiguous due to small sample sizes and insufficient control treatments and need further validation. In Limax maximus, longer photoperiods also induce the release of maturation hormones by the brain (Sokolove et al., 1981b; Sokolove et al., 1984), speeding up their growth.

Similarly, shorter photoperiods can inhibit growth and maturation and reduce reproductive rates of snails and slugs (Enée, Bonnefoy-Claudet & Gomot, 1982; Gomot, Enée & Laurent, 1982; Gomot, 1990; Hommay et al., 2001). Furthermore, a concomitant increase in temperature during short days was shown to partially compensate the effect of the shorter light phase in egg laying in Helix pomatia (Gomot, 1990). In contrast, in the slug Ambigolimax valentianus a longer photoperiod (LD 16:8) suppressed growth and reproductive maturation when compared to a 12:12 regime (Udaka & Numata, 2008). In a follow-up study Numata & Udaka (2010) showed that, while this effect of short photoperiods on maturation was the same in different populations in Japan, the effect on growth was different depending on the latitude of origin: in Osaka, the same results as before (less growth in longer photoperiods) were obtained, but in Sapporo, longer photoperiods induced growth. That led those authors to hypothesize that the two populations of this exotic slug had already diverged genetically in this trait. This result is reminiscent of the study of Lundelius & Freeman (1986) with the freshwater snail Peregriana peregra, in which a long-day or short-day response is mediated by a single gene locus.

Finally, although the majority of the studies have dealt with photoperiods, temperature could also be a reliable seasonal cue for fine tuning terrestrial gastropod reproduction like in insects and vertebrates. Thus, these animals could use temperature in addition to photoperiod for the control of reproduction, but this has not been duly tested so far (Numata & Udaka, 2010).

Hibernation.

The existence of a photoperiodic control of the annual rhythm of hibernation in stylommatophorans was first demonstrated in Helix pomatia (Jeppesen & Nygård, 1976; Jeppesen, 1977) and Cornu aspersum (Bailey, 1981). Seasonal changes in photoperiod induces physiological changes in the snails for preparation for hibernation (e.g., energy reserves, supercooling; Hunter & Popovich, 1977; Ansart, Vernon & Daguzan, 2001) and there is a threshold in the duration of the light phase that induces hibernation (Attia, 2004). Shorter photoperiods induce hibernation in the laboratory (e.g., Aupinel, 1987).

Artificially long photoperiods can make the snails terminate hibernation early or skip it altogether (Jeppesen & Nygård, 1976; Jeppesen, 1977; Bailey, 1981). Still, the emergence from hibernation is more complex and some plasticity in determining termination may be involved, with photoperiod, temperature, and relative humidity all potentially playing a role (Jeppesen & Nygård, 1976; Jeppesen, 1977; Bailey, 1981). The length of hibernation is related to latitude in Cornu aspersum, with populations displaying a shorter hibernation in lower latitudes (Iglesias, Santos & Castillejo, 1996), which is an expected result.

During hibernation, many species bury themselves in the ground or in similar hideouts, where it might be difficult to perceive environmental light/dark cycles (Wünnenberg, 1991; Wünnenberg, 1994). Thus, it has been speculated if daily physiological rhythms are maintained during hibernation, but this is still poorly understood. For instance, it is known that the daily rhythm of heart rate in Helix pomatia free-runs during hibernation (Wünnenberg, 1994).

Hibernation in many areas of the globe means that the animals must be tolerant (or at least partially so) to freezing. For Cornu aspersum, shorter photoperiods were deemed to be related to lower supercooling points and better freeze tolerance (Biannic & Daguzan, 1993). It was later shown that the shortening of the photoperiod triggers the supercooling ability via lower temperatures of crystallization (Ansart, Vernon & Daguzan, 2001). In nature, this would happen long before the onset of winter conditions, which is hypothesized to allow the species to cope both with milder and harsher winter conditions throughout its geographic range (Ansart, Vernon & Daguzan, 2001).

The supercooling ability was further investigated in Helix pomatia, in which concomitant shorter photoperiod and low temperature induced the accumulation of glycerol in preparation for the winter (Nowakowska, Caputa & Rogalska, 2006). Glycerol was considered a modest cryoprotectant in that species (as seen in the Ellobiidae Melampus bidentatus; Loomis, 1985), but glucose was deemed not to play this role (Nowakowska, Caputa & Rogalska, 2006). Unfortunately, Nowakowska, Caputa & Rogalska (2006) did not include test groups aimed to disentangle the effects of temperature and photoperiod in their study. However, as mentioned above, Udaka, Goto & Numata (2008) noted that exposure to either a shorter photoperiod or lower temperatures enhance cold tolerance in the slug Ambigolimax valentianus. Udaka, Goto & Numata (2008) also noted that acclimation to colder temperatures differs from one species of terrestrial gastropod to the next; thus, likely photoperiod is the main factor controlling this process.

Aestivation.

Aestivation is the dormancy during unfavourable environmental conditions such as dryness and summer heat. The same term is used in the literature for both short-term processes of aestivation (immediate response to adverse conditions), as well as longer processes that can include a seasonal factor such as a dry season (Heller & Dolev, 1994; Storey, 2002). Both are observed in snails and are often treated as the same thing, even though the mechanisms involved might be different. Data on the literature is consequently mixed up and even seemingly contradictory at some points. The different types of aestivation hints that it will differ not only between taxa but also from temperate regions (where most studies take place) to tropical ones.

In short-term aestivation, air and substratum temperature, alongside a stark drop in relative humidity, are the likely triggers for its start, while a return to “normal” conditions signals its end (Herreid & Rokitka, 1976; Attia, 2004). Contrary to hibernation, it is known that photoperiod plays no role in the onset of this kind of aestivation (Cook, 2001; Attia, 2004).

Long-term aestivation, however, is likely an annual event in species that undergo a dry season. For instance, the door snail Cristataria genezarethana, from Israel, starts to aestivate before the end of the rainy season, suggesting that long-term aestivation is not just a direct response to dry conditions, but likely results from physiological programming (Heller & Dolev, 1994). There are also indications that photoperiod might play a role at least in the termination of long-term aestivation in the Mediterranean snail Otala lactea (Herreid & Rokitka, 1976) and the tropical Allopeas gracile (Rahman, Mitra & Biswas, 1975). The latter authors hypothesized that a biological clock would be involved in this type of aestivation.

Other hints that long-term aestivation might be more complex were obtained from Sphincterochila candidissima, a species that both hibernates and aestivates in Spain. It was shown that individuals have a phase of feeding and storing energy before each of those events, indicating programmed preparation (Moreno-Rueda & Collantes-Martín, 2007). Similarly, the slug Ambigolimax valentianus presents heat tolerance in addition to cold tolerance (see above) and both vary seasonally (Udaka, Goto & Numata, 2008). In that species, a longer photoperiod and higher temperatures can each increase heat tolerance in a similar fashion as cold tolerance is enhanced by short-day conditions and/or lower temperatures.

It has been shown that aestivation length has an impact in other stages of the annual cycle. For instance, in the tropical snails Achatina fulica and Tanychlamys indica, a longer aestivation delays egg laying and, if long enough, can even prevent it (Raut & Ghose, 1980). This fact makes some aestivating species particularly susceptible to impacts of climate change.

There are many studies on the metabolism of aestivating snails, particularly of Otala lactea (e.g., Brooks & Storey, 1997; Storey, 2002; Ramnanan, Bell & Hughes, 2017), but there is still little data on the clock-controlled daily activity rhythms during aestivation (Attia, 2004). It is known that xerophilic species such as Eobania vermiculata and Cernuella virgata present daily rhythms of respiration and heart rate during aestivation (Kratochvil, 1976). Furthermore, it has been shown that the heart rate rhythm of Helix pomatia is kept synchronized to the environment’s photoperiod during short-term aestivation (Wünnenberg, 1991).

Other Terrestrial Gastropods

Outside the major diversification of the Stylommatophora, the remaining terrestrial gastropod lineages belong to different groups that can be summarized as (1) the non-stylommatophoran pulmonates (Orders Ellobiida and Systellommatophora), and (2) operculate land snails, including disparate lineages within the subclasses Heterobranchia, Neritimorpha and Caenogastropoda (Fig. 1).

Ellobiida

The Ellobiida include littoral species and terrestrial species. The latter (subfamily Carychiinae) comprises minute snails, including some (Zospeum spp.) that live in aphotic zones of caves.

As previously mentioned, a comparison between both Ellobiida and Systellommatophora (below) with Stylommatophora would be of great interest given their close evolutionary relationship. Among the marine Ellobiida, there are some studies available on species living in tidal environments such as the salt marsh snail Melampus bidentatus, including phenological data on tidal and lunar rhythms, and seasonality (e.g., Russel-Hunter, Apley & Hunter, 1972; Price, 1979; Price, 1984). For instance, activity of the amphibious M. bidentatus is related to both dark phase and low tide, and reproduction is, likewise, seasonal and tidal (Russel-Hunter, Apley & Hunter, 1972; Price, 1979; Price, 1984). No studies on the biological rhythms of fully terrestrial species have been conducted so far, thus, any extrapolation must be made with caution, as marine ellobiids are much different from their terrestrial counterparts.

As in pulmonate land snails and slugs, the change to longer photoperiods is thought to modulate reproduction in littoral ellobiids (Price, 1979), though the effects of light have not been fully tested free of temperature. Seasonal increase in glycerol concentrations towards the winter has been considered related to protection against freezing in M. bidentatus (Loomis, 1985) like in stylommatophorans. The haemolymph supercooling point of this species increases towards the end of winter (Loomis, 1985).

Systellommatophora

The Systellommatophora are a relatively small taxon of Eupulmonata, containing only two groups of slugs: the Onchidiidae, with both marine and terrestrial forms, and the fully terrestrial Veronicelloidea, popularly known as leatherleaf slugs.

Among the Onchidiidae, studies on the reproduction of Peronia verruculata from the Indian Ocean have hypothesized that photoperiod is a cue for maturation and egg laying (Deshpande, Nagabhushanam & Hanumante, 1980; Deshpande & Nagabhushanam, 1983). Studies on the non-ocular photoreceptors of the same species (located on the pleuro-parietal ganglia and abdominal ganglion) suggested that they might have a role in circadian photoentrainment (Gotow, 1975; Gotow & Nishi, 2009; Shimotsu et al., 2010).

Notably, a study of clock genes (per and cry1) in Onchidium reevesii (Xu, Yang & Shen, 2019) showed different results (expression patterns) from what was expected given the knowledge on Aplysia sea hares. That discrepancy puzzled those authors because they did not recognize that Onchidiidae is a lineage with a unique evolutionary history and far removed from aplysiids. This is a further reminder that taking phylogenetic relationships in consideration is essential for comparative studies; Eupulmonata will share some features (plesiomorphies) with other Heterobranchia, but we should expect many unique features (apomorphies) as well.

Daily rhythms

The biological rhythms of Veronicelloidea are scarcely studied. This is largely a pantropical group, and the dearth of studies could be a reflection of the legacy of Western colonialism and the possibilities and priorities of research in underfunded Global South countries. There are reports on the activity patterns and feeding rhythms of two species (Supplementary Material 1): Laevicaulis alte, indigenous to Africa but introduced elsewhere, and Sarasinula linguaeformis, from South America. While the former is crepuscular with a bimodal activity pattern (sunset and first hours of evening, and final hours of night and sunset; Raut & Panigrahi, 1990), the latter is nocturnal but extending its activity to the twilight hours (Junqueira et al., 2009).

Panigrahi, Mahete & Raut (1992) reported a peak in adrenalin and noradrenalin in the brain of Laevicaulis alte during the second half of the dark phase. Those authors hypothesized a link between the peak in those substances and an increase in the slugs’ activity during that phase. The lowest level in both adrenalin and noradrenalin occurs immediately before the light phase, when the animals become inactive (Panigrahi, Mahete & Raut, 1992).

Annual rhythms

There are some studies focusing on defining and describing life history traits—and the reproductive phase in particular—of veronicellid slugs, typically conducted in laboratory conditions and often regarding anthropochorous and/or invasive species (Supplementary Material 1). While seasonality is implicit in those studies, no research involving photoperiod or other cues has been conducted so far.

Operculate land snails

There is scarce data published on daily or annual rhythms of operculate land snails besides a general preference for dusk or dark times for activity. There is data on marine and freshwater Neritimorpha and the Caenogastropoda, but these clades are so old and diverse that the extent to which those finding can be extrapolated to their terrestrial relatives is uncertain.

Future Directions

In order to understand the evolution, function and significance of biological clocks and calendars we need to go back to the “real world” (Schwartz, Helm & Gerkema, 2017; Numata & Udaka, 2010). Land snails could be good candidates of non-model species to improve our understanding of clocks and calendars in the “real world”, and in this review we covered the current state of knowledge of land snail Chronobiology. In this final part we will cover the main knowledge gaps and will also highlight areas where using land snails as models may be particularly helpful in moving the Chronobiology field forward. Of course, this is in no way a comprehensive list, but we hope that it might spark new ideas and novel research lines around the world.

Neuroanatomy

The importance of detailed anatomical and physiological understanding of the study system cannot be overstated and this is an area in which further investigation is urgent. The large size and relatively small number of neurons in molluscs have made them ideal for neurobiology studies. It is to be expected that these animals would therefore also be excellent models for studying the neural mechanisms of biological rhythms in animals.

The foremost topics would arguably be determining the oscillators and the role of non-ocular photoreceptors. Non-ocular photoreceptors have been found in Stylommatophora, Systellommatophora, Hygrophila and marine Heterobranchia, so it would be interesting to uncover the evolutionary relationships of this system among these groups. For example, a starting point would be to characterize the photoreceptors of various groups and then map character states into a phylogenetic tree to assess when innovations appeared and how many times they did so (i.e., whether there is homoplasy).

Molecular pathways

While the endocrine control of reproduction is well understood in heterobranchs (e.g., Wayne, 2001; Flari & Edwards, 2003), most aspects of the pathways from Zeitgeber to rhythm expression remain uncertain, even though some headway has been made (e.g., Sokolove et al., 1984; Numata & Udaka, 2010). That is the case not only for annual rhythms like reproduction but for daily rhythms as well. Knowledge obtained from studies in freshwater Hygrophila snails (e.g., Hamanaka, Hasebe & Shiga, 2023) will undoubtedly prove important for the study of terrestrial eupulmonates, but further lineages of land snails (i.e., the operculates) will have their own idiosyncrasies. Similarly to the study of photoreceptors proposed above, a comparative phylogenetic approach of the molecular pathways may also yield interesting results given the richness of snail diversity.

Genetic makeup

Understanding the genetics of photoperiodism in animals is a vibrant and expanding field, however that effort has yet to “contaminate” the study of land snails. While much still needs to be explored, described, and tested, one immediate example of an avenue of research is the gene controlling long-day/short-day response in annual rhythms reported by Lundelius & Freeman (1986) in freshwater snails and potentially observed by Udaka & Numata (2008) in terrestrial slugs.

Nocturnality

In all marine Heterobranchia studied so far, the central clock neural rhythms are “diurnal” whereas the species’ circadian behaviours can be either nocturnal or diurnal (Block & Colwell, 2014). Stylommatophorans are primarily nocturnal, and this might be a synapomorphy of this clade. Whether this would imply differences in the functioning of the central clock in these animals compared to heterobranchs, other molluscs, and even vertebrates, remains to be determined. A broader comparison across various vertebrate and invertebrate lineages may reveal patterns that are specific to the groups and those that are general and convergent in nocturnal and diurnal species across the tree of life.

Extreme photoperiods and rhythmicity

One exciting aspect of land snails is the wide variety of environments that they colonized. Moreover, some land snail species (e.g., members of Helicidae and Achatinidae) have wide distributions through “normal” and extreme environments. Studying the rhythms of species living in environments with extreme photoperiods (e.g., arctic, caves) should help shedding light in the mechanisms of timekeeping in land snails in general, as several distinct lineages colonized these environments independently. For example, similarly to studies in drosophilids (e.g., Beauchamp et al., 2018; Helfrich-Förster, Bertolini & Menegazzi, 2018; Bertolini et al., 2019), it would be worthwhile to compare similarities and differences in molecular, physiological, and behavioural rhythms of closely related species or even populations of the same species that live in extreme and non-extreme conditions. This would elucidate which clock mechanisms allow organisms to fare in these extreme environments. Moreover, such investigation could be the first step to understand natural selection of biological clocks in nature, in which land snails can be particularly interesting models (see below).

A similar venue of investigation is whether microgastropods that are leaf litter dwellers are indeed arrhythmic in nature (as suggested by Baur & Baur, 1988), and whether they have a circadian rhythm under constant conditions in the lab.

Taxonomic coverage

Most studies were conducted in stylommatophoran species from temperate regions (e.g., nearly all studied species show long-day responses in their annual rhythms; Numata & Udaka, 2010). Thus, it would be beneficial to study species from warmer climates, particularly in the tropics, where the largest portion of biodiversity is. That is, of course, a reflection of world history, Western colonialism, and bad distribution of wealth. As we advance, greater focus (and funding) must be put into the Global South.

Nevertheless, while many species are expected to have their idiosyncrasies, all Stylommatophora (and perhaps all Eupulmonata) are expected to largely share the same system controlling their rhythmicity. Therefore, coverage of non-stylommatophoran taxa is urgent. Operculate snails belong to lineages as far from stylommatophorans as we are from the other main vertebrate branches; so, despite being called “land snails” they are completely distinct organisms, with their own evolutionary history and thus, interesting to be studied on their own. Furthermore, the paraphyletic nature of “terrestrial gastropods” provides a good opportunity to identify the relative importance and weight of evolutionary heritage vis-à-vis adaptations to the terrestrial environment.

Aestivation

Snails are capable of both hibernation and aestivation. In general, aestivation is at the same time thought to be simpler than hibernation (Cook, 2001; Attia, 2004), but still not that well understood. As mentioned above, there are two types of aestivation: short-term and long-term. Based on the information presently available, short-term aestivation is likely a direct response to unfavourable environmental conditions , while long-term aestivation seems to have a programmed aspect to it (e.g., Rahman, Mitra & Biswas, 1975; Herreid & Rokitka, 1976; Heller & Dolev, 1994; Moreno-Rueda & Collantes-Martín, 2007). For species living in regions with marked alternance of rainy and dry seasons (e.g., arid environments, Mediterranean and monsoon climates, and various other tropical settings), it is intuitive to hypothesise that their aestivation includes physiologically programmed processes.

Host-parasite interactions

Snails are hosts to a number of parasites including those that are able to modify their host behaviour (e.g., “zombie snails”). It is known that parasites can modulate their host’s rhythms (Carvalho Cabral, Olivier & Cermakian, 2019), but the effects of parasites on the circadian rhythms of molluscs remain virtually unexplored. It has been shown that trematodes can alter activity patterns of the pond snail Lymnaea stagnalis, increasing foraging events when potential predators (definitive hosts) are present (Voutilainen, 2010). Similar results have been observed in non-heterobranch freshwater snails (e.g., Levri et al., 2007). Nevertheless, these freshwater snails are diurnal, so it can be expected that nocturnal land snails and slugs would be more affected by parasites dependant on diurnal predators.

Natural selection

Biological clocks are considered essential to the survival of organisms, with lab studies showing negative selection on clocks that differ too much from the period of the environmental cycle (Ouyang et al., 1998; Horn et al., 2019), as well as genetic studies proposing local adaptation of clocks (Kyriacou et al., 2008; Hut et al., 2013; Helfrich-Förster, Bertolini & Menegazzi, 2018). Apart from populational differences, it is also interesting to note that properties of circadian clocks vary between individuals of the same species (e.g., tau in Cornu aspersum; Lorvelec et al., 1991), are genetically determined (e.g., long-day and short-day alleles of Peregrina peregra; Lundelius & Freeman, 1986), and heritable (Helm & Visser, 2010). Given that, it would be expected that natural variation of clocks would lead to individual differences in physiology and/or behaviour in nature, which, in turn, could lead to fitness differences depending on the environment. These topics are still understudied in natural populations and land snails could be very interesting model systems. Apart from the comparative studies described above, classical experiments such as common garden studies and reciprocal translocations would allow more direct investigations of local adaptation and selection (De Villemereuil et al., 2016). Finally, studies of animals in nature are important when measuring selection, because selective pressures only make sense when measured in the real world (Helm et al., 2017; Schwartz, Helm & Gerkema, 2017). It is very challenging to carry out studies like reciprocal translocations in the wild with many of the traditional model systems, but it should be possible with land snails, especially the larger-bodied species.

Global change

Anthropic changes in land use and the climate are generating an array of issues that can be studied under a biological rhythms’ perspective. Aridification and increasing temperatures will lead to local extinctions, range shifts and/or migrations. For instance, higher temperatures can increase the time snails must aestivate and thus shorten the phase of growth and feeding, which hypothetically can lead to local extinctions (Moreno-Rueda & Collantes-Martín, 2007). In contrast, some populations might shift their ranges to areas that suddenly have more suitable climates (including shorter winters), typically migrating to higher latitudes. Migrating polewards, however, might put the snails in contact with very different photoperiods, such as those in northern Fennoscandia. It is hypothesized that these photoperiods can act as selective barriers to some species (Huffeldt, 2020). One interesting line of research involves understanding how much of what we see in temperature-related traits is evolutionary adaptation or phenotypic plasticity (Schilthuizen & Kellermann, 2014).

Invasion biology

Even though the life cycles of several invasive species have been studied in detail, the interplay between their circannual clocks and their “aptitude” in becoming invasive has not been investigated. For instance, South (1989), studying the invasive slug Deroceras reticulatum, hypothesized that a less strict annual schedule (in comparison to other slugs and snails) could be a central portion of such aptitude.

Light pollution

Another urgent topic for study is understanding if and how light pollution affects snail and slug populations. Very few studies have dealt directly with this subject so far (for a review see Hussein et al., 2021), particularly regarding terrestrial gastropods. For instance, sites exposed to artificial light at night (ALAN) were shown to attract Arionidae slugs (Van Grunsven et al., 2018), presumably the anthropochorous species though the authors did not identify their specimens to species level. No studies so far have focused on if and how ALAN affects the rhythms and biological clocks of land snails and slugs. As observed in other animals, including invertebrates, ALAN can impact daily and annual rhythms, survival, and fitness (Gaston et al., 2013; Dominoni, Borniger & Nelson, 2016; Davies & Smyth, 2018). Case in point, ALAN has been recently shown to delay development and hatching of eggs of the pond snail Lymnaea stagnalis, while increasing feeding and growth rate (Baz et al., 2022); it was also hypothesized to impact learning and memory formation in this species (Hussein et al., 2020). The longer light phase resulting from ALAN is expected to affect terrestrial gastropods as well. Understanding the effects of different wavelengths of light is also crucial (Hussein et al., 2021), as this will enable us to make better and sustainable use of artificial light.

Concluding Remarks

It has been argued that the heyday of chronobiological studies on molluscs were the 1960s to 1980s. The drop in interest starting in the mid-1980s has been assigned to novel tools that allowed studies in mammals, instead of having to rely on invertebrate model species (Wayne, 2001; Block & Colwell, 2014). That slowdown continues to this day.

We hope that our contribution will renew interest in this area and spark novel projects. As we have tried to argue above, land snails and slugs would not only make good models for chronobiology and neuroscience (Lyons, 2011) but also for evolutionary ecology. New studies would have implications that go even beyond: conservation, invasion biology, biogeography, public health, and urban ecology, to name a few.

Supplemental Information

Supplemental Information 1 List of studies on daily and seasonal rhythms of pulmonate terrestrial gastropods

Studies involving the daily (Table S1) and seasonal (Table S2) rhythms of pulmonate terrestrial gastropods (Stylommatophora and Systellommatophora). Species names have been updated according to current classification and taxonomic usage (MolluscaBase, 2023), so the names used here might differ from those in the original articles.

Supplemental Information 2 Biological rhythms of marine and freshwater Heterobranchia

Supplemental Information 3 Snail CC-BY-NC-SA

We are grateful to the staff of the UiT library for keeping up with our incessant request for papers and books from all over the world and to all the colleagues who shared with us some hard-to-find literature: Gonzalo Collado, Suzete R. Gomes, Jamen U. Otani, Barna Páll-Gergely, Christian Selbach. We also thank Daniel C. Cavallari for his comments and suggestions on an earlier version of the manuscript, and to the three anonymous reviewers for their thoughtful input.

Additional Information and Declarations

Competing Interests

Author Contributions

Data Availability

The authors declare there are no competing interests.

Rodrigo B. Salvador conceived and designed the experiments, performed the experiments, analyzed the data, prepared figures and/or tables, authored or reviewed drafts of the article, and approved the final draft.

Barbara Mizumo Tomotani conceived and designed the experiments, analyzed the data, prepared figures and/or tables, authored or reviewed drafts of the article, and approved the final draft.

The following information was supplied regarding data availability:

This is a literature review.

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
