# Peer review of "Clocks at a snail pace: biological rhythms in terrestrial gastropods"

_PeerJ, doi:10.7717/peerj.18318_

## Round 0.1 · original submission · Major Revisions

We have received three reviews for this manuscript. The reviewers had divergent opinions regarding the relevance of the proposed review. The main criticism raised is that, in its current version, the review is too superficial regarding: 1-the content of individual studies cited, 2-the discussion of general patterns that can be drawn from the previous research and 3- the suggested future directions of research in the field. This is critical since those three points should all be major aims of a review paper.

Also as suggested by reviewer 1 (and 2), contrasting terrestrial to aquatic gastropods to look for similarities and differences between groups would be useful. The manuscript also needs to clarify several other important aspects such as its main objectives and the terminology used throughout the text. Reviewers made useful comments on how to achieve this.

Reviewer 1 ·

Basic reporting

All criteria are fulfilled.

Experimental design

All criteria are fulfilled.

Validity of the findings

Criteria partially fulfilled. The manuscript stays superficial in several instances and can be strongly improved by being more specific with regard to the content of individual cited studies and the description of future research directions. See "Additional comments" for further details.

Additional comments

The authors provide an extensive overview of rhythm research in terrestrial gastropods, while highlighting important knowledge gaps and discussion directions for future research. The manuscript is well structured and I appreciate the throughout description of the literature review procedure. While the covered topic (terrestrial gastropod chronobiology) is somewhat narrow, it is nevertheless relevant for gastropod researchers in general and will also be of interest to chronobiologists work with other mollusks or invertebrates, thus fitting the scope of PeerJ well.

My only mayor critique is that currently the manuscripts stays somewhat superficial with respect to the content of individual studies as well as the discussion of possible general patterns that can be identified from the existing research. Such cases, together with other, minor aspects, are listed in the comments below. In sum, I consider the required revisions as minor, but I encourage the authors to closely reassess the complete manuscript considering the points above. This will also make it more engaging for the reader.

Further comments by line:

title The manuscript does describe (endogenous) rhythms and the environmental factors influencing them extensively, but it hardly mentions actual clocks, i.e. the mechanisms evoking endogenous rhythms. This can largely be explained by the current lack of knowledge regarding mechanistic, molecular and neurological details, but highlighting clocks this prominently in the title is therefore somewhat misleading. The title should therefore be adjusted, focusing more on the rhythm aspect.

50 The term “photoperiod” (i.e. hours of light per day) should be defined for the non-chronobiological audience.

69 The “qualities that make them exciting models” should be explicitly stated. The information in the previous paragraph similarly applies to other animal groups. What characteristics (with regard to chronobiology) set terrestrial gastropods apart from e.g. model such as mammals, birds or insects, or from aquatic gastropods? What questions can be answered with this group that cannot be answered with others?
In generally, I recommend contrasting terrestrial to aquatic gastropods for which chronobiological information is available to look for commonalities and differences between groups (e.g. towards the end of the manuscript). Expanding this to other related groups like e.g. bivalves could further help to identify the aspects that make terrestrial gastropods special and how these may relate to evolution. Molecular information is available for a number of aquatic gastropods (doi: 10.1177/074873002237136, doi: 10.1086/698467, doi: 10.1016/j.cbpa.2018.05.002, doi: 10.3390/genes10070488, doi: 10.1016/j.jphotobiol.2020.111994, doi: 10.1038/s41598-019-44526-3, doi: 10.3389/fevo.2022.1078234, doi: 10.1186/1742-9994-9-9) and a recent review includes an overview for bivalves (doi: 10.1146/annurev-marine-030422-113038).

182-3 Later in the manuscript, the term “entrainment” is used, but it is never explained. For a non-chronobiological audience, it would make sense to define the term here.

188 I would make sense to state that this applies especially to the natural environment where environmental cycles preventing a true free-run are typically ubiquitous.

198 “Activity rhythms” is not precise as e.g. metabolism or gene expression can show daily activity changes as well. Better use “locomotor activity”, “behavior” or similar.

212 Crepuscular activity is commonly defined as being activity ONLY around dusk and/or dawn, i.e. it is not used for nocturnal species even if their activity extends to the twilight hours. In fact, crepuscular animals being bimodal (i.e. active at both dusk and dawn) is the more common than being active during one of the phases.

238-40 What is the reason/benefit for the diurnal activity in this species? What is special about its lifestyle or environment?

251-6 Here it would help to explain the significance of the observed patterns. How do they relate to the lifestyle of the species or of terrestrial gastropods in general? How do they compare to other terrestrial invertebrates (e.g. insects) or aquatic gastropods? Again, what is special about terrestrial gastropods and what may be the reason for this?

258-64 What is the function of 5-methoxytryptophol (in other species) and how does it relate to melatonin? To appreciate the findings, readers have to be readily understandable.

288-9 The findings in the nocturnal Limax maximus contradicts Aschoff’s rule that states that increasing intensities of constant light cause circadian period shortening in diurnal species and a lengthening in nocturnal species. This is thus a very interesting finding and should be pointed out. I may also be worth checking, if similar observations exist in other gastropods or mollusks.

340-2 It sounds as if the Heterobranchia clock is located ONLY in the eye and nowhere else in the neural system. This would be very surprising and I struggle to believe it. Multiple oscillators in different brain regions or organs are common in many species. This should be explained in more detail instead of just referring to the supplementary material.

357-9 The statement that “extraocular receptors have a different sensitivity to light wavelengths” is too general, considering it is based on two species. It is well possible that both inside and outside their eyes there are several light receptors with different spectral sensitivities, so this should be described in more detail. Also, considering that many invertebrates entrain primarily via non-visual blue-light receptors, missing entrainment by wavelengths >600 nm in eyeless animals does not necessarily provide much information. If there is information on a greater relevance of eyes and/or longer (e.g. red) wavelengths for entrainment in terrestrial gastropods, this should be pointed out as a contrast to other groups and could be compare to marine species that primarily experience blue wavelengths.

361-3 Provide details on what this hypothesis is based on. See also comment to line 357-9.

365 At this point it would be worth adding a paragraph addressing the current lack of molecular knowledge about clocks in terrestrial gastropods (e.g. clock gene expression patterns, gene protein-interactions, identification of “clock neurons” via staining techniques, etc.). This aspect is included in the future directions at the end of the manuscript, so it makes sense to highlight this knowledge gap as well as the importance of mechanistic understanding.

378 Change to something like “Endogenous reproductive cycles have been described in several slugs and snails…”. You cannot generalize based on studies in individual species.

381 How much shorter are the cycles? What is the range? Be specific.

382 If this was the first study to describe circannual rhythms in mollusks, this is an important step and deserves some details on the work, also to raise more interest in the reader.

398-9 In which regard are the results ambiguous? Provide details.

433-5 Making a generalized statement based on a single study is not appropriate. One can make assumption based e.g. on latitudinal patterns in other animal groups, but they should be presented as such.

538-42 See comment on line 212.

595 Unclear. What kind of ocular rhythm? Light sensitivity? Pigment aggregation/dispersion? Also, instead of “diurnally expressed” better write “peak during the day” for clarity.

597 There are numerous nocturnal vertebrate species. A comparative approach with diurnal and nocturnal species from both groups could identify general and gastropod-specific adaptations.

599-601 Very vague! How exactly will these studies help understanding the evolution of terrestrial gastropod clocks? Provide e.g. suggestions for study designs.

613-6 Again, I encourage comparisons to aquatic species. The paraphyletic nature of terrestrial gastropods actually provides a good opportunity to identify the relative importance of evolutionary heritage vs. adaptation to the terrestrial lifestyle.

635-7 Too vague. Provide clear suggestion son how to study selection for clock attributes. There are several studies that look at population-specific clock gene differences or fitness/selection related to matching external (light) cycles in insects (especially drosophilids), which could serve as a template (e.g. doi: 10.3389/fphys.2019.01374, doi: 10.1111/ejn.14180).

657 The paper “Hussein et al. 2021” does not occur in the reference list. Is this maybe a typo? Also, the reference list states 2018 as publication year for another Hussein at al. paper, which was however published in 2020. Please check all references for completeness and correctness.

658 Define the term “ALAN” here (and not in line 662).

Fig.1 I recommend using a different color for the Heterobranchia group. The blue is very dark and hard to see on the black background. Also, why not just use a white background and black tree branches?

Reviewer 2 ·

Basic reporting

This review widely covers biological rhythms in terrestrial gastropods. Recently Hamanaka Y, Hasebe M, Shiga S. Neural mechanism of circadian clock-based photoperiodism in insects and snails. J Comp Physiol A Neuroethol Sens Neural Behav Physiol. 2023 Aug 18. doi: 10.1007/s00359-023-01662-6. Epub ahead of print. PMID: 37596422. has been published. The review by Hamanaka includes snails both terrestrial and water ones especially about behavior and physiological mechanism. This review includes ecological perspective and this makes a different points of view. However, I did not see clear objectives of this review. Why do you have to focus on the terrestrial snails? What are the interesting points or strength of the terrestrial molluscan study? What is the motivation of this review?

Experimental design

I think no problem in methodology.
In line 131-135, you need some references.
Line 318, paper by Ford and Cook 1988 does not use skeleton photoperiod. Please check.
About paragraph organization: Description of each theme was separated by taxonomic groups: stylommatophora and others. But I do not understand the reason for the separation. As the authors mention, in the terrestrial snails and slugs knowledge on the biological rhythm is scarce. If you keep this style you need to describe the reason. For me it would be nice to compare knowledge across the groups in the same section.

Validity of the findings

I did not see clear goals of this review and therefore conclusion is weak.
This review introduce past research carefully but I did not find developed arguments or opnion by integration of the past research.

Additional comments

The author prepared only 1 figure of phylogenetic position of land snails and slugs. But I think this does not signify this review contents. Figure 1 helps to understand the evolutionary position of the animals focused. But this does not match the title of the manuscript.
Line 339 and after: title ‘the clock’ does not fit the contents here. This section contain mainly photoreceptor for the circadian entrainment. Authors should clearly discriminate photoreceptors and clock. these two are different.
Line 387: ‘both photoperiodic and endocrine control mechanisms’ I feel uncomfortable with this expression because photoperiodic control employ endocrine mechanism.

Reviewer 3 ·

Basic reporting

no comment

Experimental design

1. Common circadian rhythm terms (e.g. τ, free-running cycle, masking, Zeitgber, entrainment) are described in a scattered manner. It would be helpful to first create a section on these terms and outline them, along with the detection of periodicity and the nature of endogenous clocks.

2. Circadian rhythms are endogenous periodicities driven by the circadian clock. Circadian rhythms exhibit autonomous oscillations, the key properties of which are entrainment and temperature compensation. The authors described the former but not the latter. The definitions of circadian clock and circadian rhythm should be clearly described and it should be pointed out whether circadian clock studies with a strict definition have been conducted in terrestrial molluscs.

Validity of the findings

no comment

Additional comments

Some changes need to be made to the description of chronobiology in this MS as it is ambiguous and unexplained and will not be helpful to chronobiologists of other organisms or readers who are not familiar with this research field.

1. in section 4.1.1. Activity rhythms:
The word "activity" is used a lot in this section, but it does not tell us anything specific about the observed behaviour, except for "feeding". If the article is about observations of locomotor rhythms, I recommend that it is described as "Locomotor rhythms" rather than "Activity rhythms".

2. In section "4.2 Seasonal rhythms", circannual rhythms are periodicities experimentally shown to be produced by circannual clocks. The property of responding to photoperiods (photoperiodicity) and circannual rhythms are two different phenomena. However, the manuscript appears to conflate the two. The authors do not use the term "photoperidoic reponse" at all. I think "seasonal rhythm" is an ambiguous term and confuses the reader. I recommend describing circannual rhythm and photoperiodicity separately.

---

## Round 0.2 · accepted · Accept

The final revisions are satisfying

Reviewer 3 ·

Basic reporting

no comment

Experimental design

no comment

Validity of the findings

no comment